# A Novel Approach for Group Decision Making Based on the Best–Worst Method (G-BWM): Application to Supply Chain Management

**Gholamreza Haseli [1]** , **Reza Sheikh [1]** , **Jianqiang Wang [2]** , **Hana Tomaskova [3],*** **and Erfan Babaee Tirkolaee [4]**

1   School of Industrial Engineering and Management Sciences, Shahrood University of Technology, Shahrood 3619995161, Iran; ghr.haseli@gmail.com (G.H.); resheikh@shahroodut.ac.ir (R.S.)
2   School of Business, Central South University, Changsha 410083, China; jqwang@csu.edu.cn
3   Faculty of Informatics and Management, University of Hradec Kralove, 50003 Hradec Kralove, Czech Republic
4   Department of Industrial Engineering, Istinye University, Istanbul 34010, Turkey; erfan.babaee@istinye.edu.tr
*   Correspondence: hana.tomaskova@uhk.cz

**Abstract:** Due to the complexity of real-world multi-criteria decision-making (MCDM) issues, analyzing different opinions from a group of decision makers needs to ensure appropriate decision making. The group decision-making methods collect preferences of the decision makers and present the best preferences using mathematical equations. The best–worst method (BWM) is one of the recently introduced MCDM methods that requires fewer pairwise comparisons to obtain the criteria weights than the other MCDM methods. In this research, we develop a novel approach to group decision-making problems based on the BWM called G-BWM. This approach helps us to analyze the preferences of decision makers to carry out democratic decision making using the BWM structure. In order to assess the applicability of the proposed methodology and represent its novelty, two numerical examples from the literature with the application to supply chain management (SCM) (i.e., green supplier selection and supplier development/segmentation) are examined and discussed. The results demonstrate the performance of our proposed G-BWM for group decision making in terms of a large number of decision makers, ease of use and achieving democratic decisions in the decision-making process.

**Keywords:** multi-criteria decision making; group decision making; best–worst method; G-BWM; supply chain management

## 1. Introduction

Decision making can be considered as the choice of the best alternative among a set of alternatives according to a number of effective criteria [1]. Typically, decision making for real-world problems is complicated, and it is impossible to reach the expected decisions with just one effective criterion [2,3]. In the case of multiple criteria to solve a decision-making problem, the implementation of multi-criteria decision-making (MCDM) methods is recommended [4]. In fact, MCDM methods are utilized in different scientific fields such as computer science [5,6], service quality [7,8], supply chain management (SCM) [9–11], engineering [12,13], health/medicine [14–16], etc.

MCDM mainly includes two sections when dealing with scientific problems. First, it determines the information of decisions, such as the criteria weights, and second, it collects the criteria information and ranks the alternatives based on this information [4]. Over recent years, several MCDM methods, such as the analytic hierarchy process (AHP) [17], analytic network process (ANP) [18], step-wise weight assessment ratio analysis (SWARA) [19], and base-criterion method (BCM) [20], have been suggested to acquire the criteria weights.

Rezaei [21] believes using the unstructured approach in executing the pairwise comparisons is the main reason for the inconsistency. The introduction of the best–worst method

(BWM) improves the consistency ratio by performing fewer pairwise comparisons [21,22]. BWM is easy and precise because the implementation of secondary comparisons is not necessary [23,24]. A review of the latest research works in the MCDM problem field shows that the BWM has been utilized successfully by researchers. Researchers applied BWM to make decisions in the different MCDM problems, such as sustainability assessment [25–27], supplier selection [28,29], risk evaluation [30–32], airport-related evaluation [33–35], efficiency measurement [36,37], selection of location and equipment [38,39], urban transportation network evaluation [40,41], etc.

By increasing the complexity of decisions in modern environments, for an individual decision maker, it is difficult to make an optimal decision by considering the aspects of the problem [42]. Moving from decision making as an individual to groups of decisio makers leads to complexity in the analysis of decision makers' opinions [43]. Hafezalkotob and Hafezalkotob [44] offered a group decision-making approach based on BWM in order to support the group decision-making process. They tried to help the senior decision maker take into account both democratic and autocratic styles. They tested the applicability of their proposed method on two case study problems. Furthermore, Safarzadeh et al. [45] extended the BWM through a novel approach to the group decision-making method. The proposed approach contains three steps and M1 and M2 mathematical algorithms to obtain the criteria weights.

It should be noted that the developed models for group decision making based on BWM have some difficulties that can restrict their applications. Given the increase in the number of decision makers on the expert panel, the size of the mathematical model also grows. Furthermore, sometimes all decision makers are on an equal level and have equal influence on decision making. In this case, we cannot choose the senior decision maker to deal with the decision making. Moreover, there may be significant conflict among decision makers' opinions. In these situations, the proposed approaches try to eliminate the inconsistent opinions of decision makers who have minorities. Therefore, it is necessary to build up a new approach to eliminate the weaknesses of previously developed models for group decision making based on BWM.

This study aims to illustrate how our proposed group decision-making BWM (G-BWM) can be implemented when a large number of decision makers are taken into account. We also show how different decision makers with various degrees of importance can be categorized for optimal analysis. Furthermore, it is demonstrated how the optimal weights of criteria are obtained without eliminating the opinions of decision makers who have minorities.

The remainder of this paper is organized as follows. Section 2 introduces the steps of BWM and describes the suggested G-BWM. In Section 3, two numerical examples of SCM are presented and investigated. Section 4 provides a comparative analysis and discussion on the advantages of G-BWM, and, finally, in Section 5, the conclusions and recommendations for future research are given.

## 2. G-BWM

Assume that there are $n$ criteria in the decision-making problem. A comparison of the relative importance of the criteria can be executed using the scale of $\langle 1, 9 \rangle$. The resulting pairwise comparison matrix is as follows:

$$A = \begin{bmatrix} a_{11} & a_{12} & \cdots & a_{1n} \\ a_{21} & a_{22} & \cdots & a_{2n} \\ \vdots & \vdots & \ddots & \vdots \\ a_{n1} & a_{n2} & \cdots & a_{nn} \end{bmatrix}, \tag{1}$$

where $a_{ij}$ shows the relative preference of criterion $i$ over criterion $j$. Here, $a_{ij} = 1$ stands for the equal relative preference between criterion $i$ and criterion $j$. Similarly, $a_{ji}$ represents the relative preference of criterion $j$ over criterion $i$, which can be written reciprocally

($a_{ij} = 1/a_{ji}$). If $a_{ij} > 0$, criterion $i$ is of importance over criterion $j$ and $a_{ij} = 9$ denotes the extreme relative preference of criterion $i$ over criterion $j$.

MCDM methods such as AHP require $n(n-1)/2$ pairwise comparisons to obtain the criteria weights using the pairwise comparison matrix [46]. Rezaei [21] introduced the BWM and divided the steps of pairwise comparisons into two parts: (i) reference comparison and (ii) secondary comparisons. He could reduce the required number of pairwise comparisons to $2n - 3$ ($n - 2$ pairwise comparisons of best criteria to other criteria $+n - 2$ pairwise comparisons of other criteria to the worst criterion $+1$ pairwise comparisons of the best criterion to the worst criterion) [4].

The group decision-making method based on BWM introduced by Hafezalkotob and Hafezalkotob [44] adheres to the principles of reference and secondary comparisons and simultaneously supports the opinions of $k$ decision makers (decision makers from the expert panel) and a senior decision maker. A senior decision maker can evaluate the importance and expertise of each decision maker based on their skill, talent, and knowledge. Regarding the disadvantages raised in the first section of this paper, our novel G-BWM is proposed for situations in which there is no senior decision maker.

Suppose that there are many decision makers to solve a decision-making problem. By increasing the panel size of decision makers, the scale of mathematical models also increases. Moreover, the evaluations made by decision makers to select the best and worst criteria may be different. To prevent increasing the complexity of the mathematical model, the decision makers can be categorized based on their evaluations of the best and worst criteria. Grouping decision makers whose evaluations are similar to each other makes the calculations and analyses easier. Suppose that there are 10 decision makers to decide on a set of criteria. Therefore, the decision makers are divided into 3 groups based on their evaluations (Figure 1).

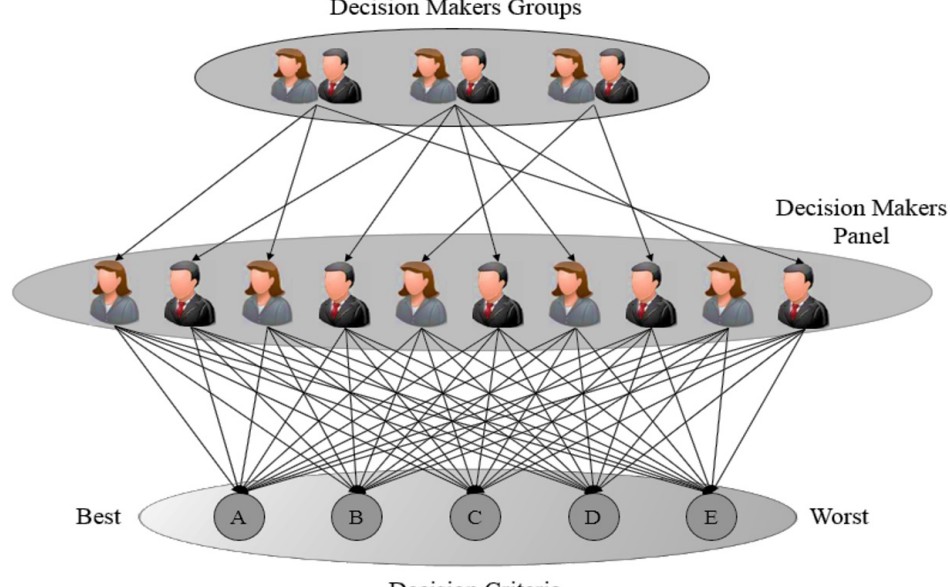

**Figure 1.** Decision maker groups.

Furthermore, the number of decision makers may be different in the various groups (Figure 2). Decision makers in the same group choose the same criteria as the best and worst criteria. The relative importance of decision makers' evaluations for pairwise comparisons will be matched using the "geometric mean". In fact, rather than performing analysis for evaluations of each decision maker, the analysis is performed for each group. Accordingly, the number of analytical steps is reduced to obtain the criteria weights in this approach.

In the geometric mean, a set of numbers are multiplied and then the $n$ th root is obtained, where $n$ stands for the count of numbers in the given set [47] Taking the geometric

mean is known as one of the best methods in group decision-making problems. The geometric mean applies to positive numbers because it takes the $n$th root [48]. Given that the BWM employs a positive scale of $\langle 1, 9 \rangle$ to execute pairwise comparisons, the geometric mean is a suitable method to indicate the typical value for pairwise comparisons.

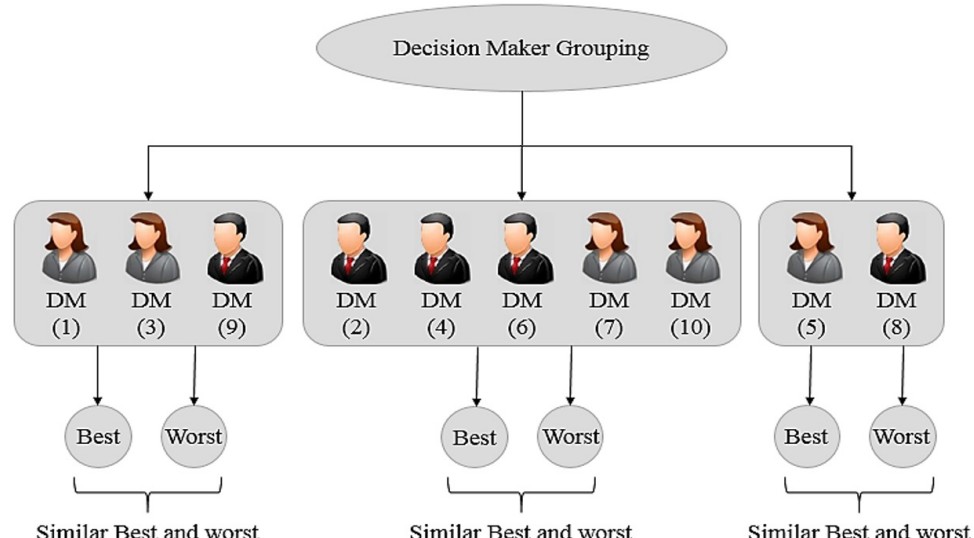

**Figure 2.** Best and worst criteria in each group.

### 2.1. Steps of G-BWM

Step 1. Determine the decision makers and decision criteria.

We consider a set of criteria $\{C_1, C_2, \cdots, C_m\}$ to achieve a decision through our decision makers $\{DM_1, DM_2, \cdots, DM_n\}$.

Step 2. Determine the most important (best) and the least important (worst) criteria.

Each decision maker selects the best and the worst criteria in general.

Step 3. Perform the pairwise comparisons of the best over other criteria using the crisp numbers of 1 to 9. The crisp numbers of 1 to 9 are numerical scales presented to determine the relative importance of pairwise comparisons. Here, $a_{ij} = 1$ represents the equal importance of the criterion $i$ and over criterion $j$. Moreover, $a_{ij} = 9$ stands for the extreme importance preference of criterion $i$ over criterion $j$. The vector of the best criterion over other criteria would be:

$$A_{Bj} = (a_{B1}, a_{B2}, \cdots, a_{Bm}) \qquad (j = 1, 2, 3, \cdots, m) \, , \qquad (2)$$

where $a_{Bj}$ denotes the relative importance value of the best criterion over criterion $j$.

Step 4. Perform the pairwise comparisons of all criteria over the worst criterion using the crisp numbers of 1 to 9. The vector of all criteria over the worst criterion would be:

$$A_{jW} = (a_{1W}, a_{2W}, \cdots, a_{mW}) \qquad (j = 1, 2, 3, \cdots, m) \, , \qquad (3)$$

where $a_{jw}$ stands for the relative importance value of criterion $j$ over the worst criterion.

Step 5. Group decision makers based on their choices about the best and worst criteria.

The decision makers who choose the same criteria as the best and worst fall into one group $G_i$, where $i = 1, 2, \ldots k$ and $k$ is the number of groups. The result of grouping decision makers would be:

$$Group_{DM} = (G_1, G_2, \cdots, G_k) \, . \qquad (4)$$

Step 6. Take the geometric mean using the total preference of the best criterion over other criteria (Total $A_{Bj}$) and total preference of all criteria over the worst criterion (Total $A_{jw}$) for each group. In this step, evaluations of the decision makers are calculated

for each Total $A_{Bj}$ and Total $A_{jw}$ using the geometric mean within each group. For each group $(G_1, G_2, \cdots, G_k)$:

$$
\begin{aligned}
\left(\prod_{i=1}^{n} \text{Total } a_{Bj(DM_i)}\right)^{\frac{1}{n}} &= \sqrt[n]{a_{Bj(DM_1)} \times a_{Bj(DM_2)} \times \cdots a_{Bj(DM_n)}}\ , \\
\left(\prod_{i=1}^{n} \text{Total } a_{jW(DM_i)}\right)^{\frac{1}{n}} &= \sqrt[n]{a_{jW(DM_1)} \times a_{jW(DM_2)} \times \cdots a_{jW(DM_n)}}\ .
\end{aligned}
\tag{5}
$$

Step 7. Obtain the optimal value of criteria weights $(w_1, w_2, \cdots, w_n)$ for each group.

The optimal values of weights for $w_B/w_j$ and $w_j/w_W$ are equal to $a_{Bj}$ and $a_{jW}$, respectively. Since the criteria weights are aggregated and non-negative, the mathematical model can be written as follows:

$$
\begin{aligned}
& \text{minimize } \max_{j} \left|\frac{w_B}{w_j} - a_{Bj}\right|, \left|\frac{w_j}{w_W} - a_{jW}\right| \\
& \text{subject to } \begin{cases} \sum_{j=1}^{n}(w_j) = 1 \\ w_j \geq 0 \ \text{ for all } j \end{cases}
\end{aligned}
\tag{6}
$$

Now, Model (6) can be written as follows:

$$
\begin{aligned}
& \text{minimize } \xi \\
& \text{subject to } \begin{cases} \left|\frac{w_B}{w_j} - a_{Bj}\right| \leq \xi, \\ \left|\frac{w_j}{w_W} - a_{jW}\right| \leq \xi, \\ \sum_{j=1}^{n}(w_j) = 1, \\ w_j \geq 0 \ \text{for all } j. \end{cases}
\end{aligned}
\tag{7}
$$

The optimal value of criteria weights $(w_{n1}, w_{n2}, \cdots, w_{nn})$ for each group as well as the value of $\xi$ can be determined by solving Model (7). According to Model (7), the total weight of the criteria must be equal to 1. Each of the criteria that receives a higher weight value than the other criteria is of a higher priority.

Step 8. Calculate the final weights using the average weights obtained for each group. The optimal value of weight obtained for each criterion in each group is multiplied by the number of decision makers within that group, and then the sum of the results is divided by the number of decision makers.

$$
w_j = \frac{\sum_{k=1}^{n}(w_{jk} \times n_k)}{N} \qquad \forall j,
\tag{8}
$$

where $n_k$ represents the number of decision makers in the $k$th group and $N$ shows the total number of decision makers where $N = (n_1, n_2, \cdots, n_n)$. The various steps of implementing the proposed G-BWM are also illustrated as a flowchart in Figure 3.

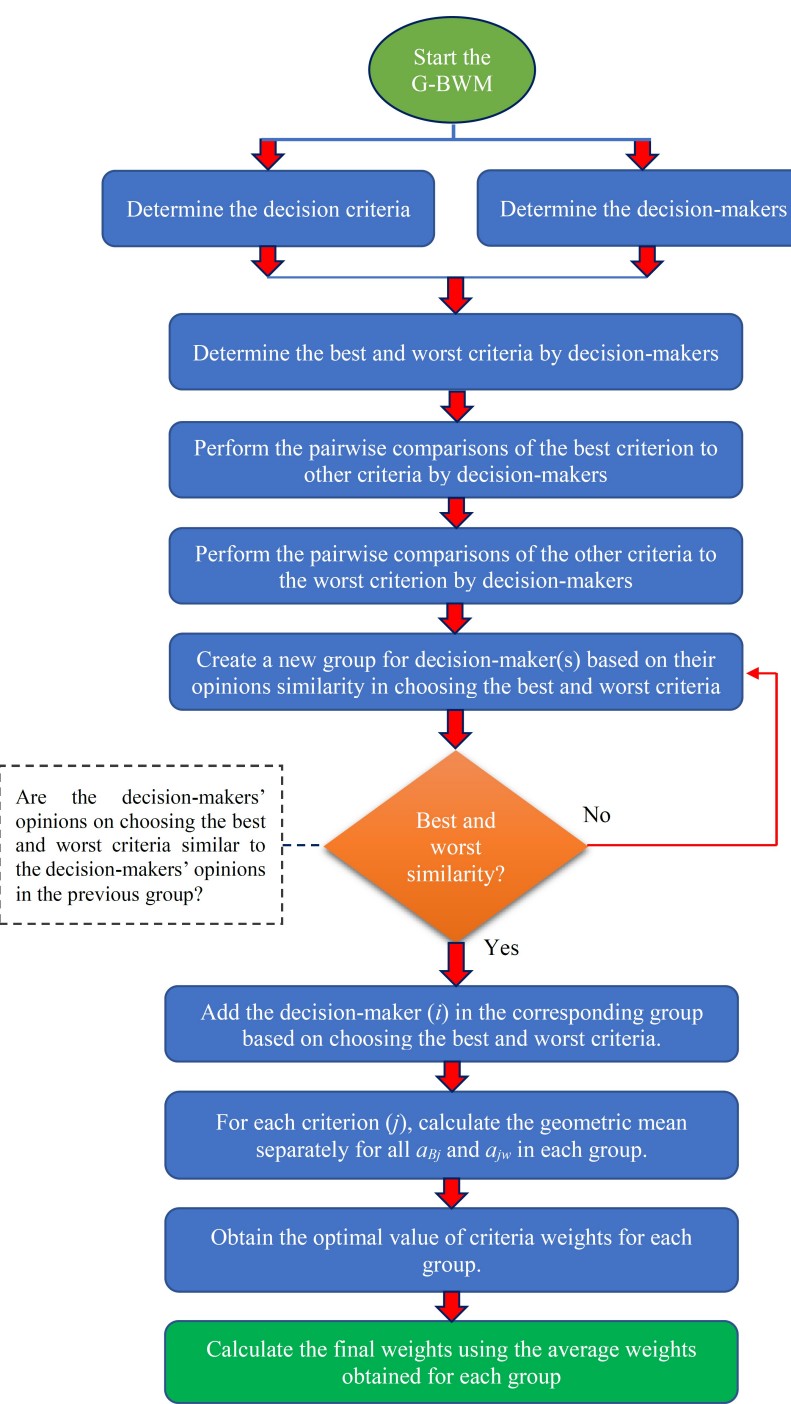

**Figure 3.** Flowchart of the proposed G-BWM.

### 2.2. Consistency Ratio

The pairwise comparisons are fully consistent when $a_{Bj} \times a_{jw} = a_{Bw}$. The consistency ratio is computed using $\xi$ and the consistency index value. The consistency ratio is an indicator for the consistent degree of comparisons. However, pairwise comparisons may not be fully consistent. In other words, the comparisons become less reliable for larger values of $\xi$. By solving Model (7) for different values of $a_{BW} \in \{1, 2, \cdots, 9\}$, the maximum possible $\xi$ can be found. Moreover, the values listed in Table 1 are employed as the consistency index [21].

**Table 1.** Consistency index (CI) [21].

| $a_B W$ | 1 | 2 | 3 | 4 | 5 | 6 | 7 | 8 | 9 |
|---|---|---|---|---|---|---|---|---|---|
| Consistency Index | 0 | 0.44 | 1 | 1.63 | 2.3 | 3 | 3.73 | 4.47 | 5.23 |

Now, the consistency ratio can be computed as follows:

$$\text{Consistency Ratio} = \frac{\xi}{\text{Consistency Index}} . \tag{9}$$

With regard to Table 1, the maximum possible value of $a_{Bw}$ is 9. When $a_{Bj} = a_{jw} \neq a_{Bw}$, the inconsistency of pairwise comparison occurs, which means that $a_{Bj} \times a_{jw}$ may be lower or higher than $a_{Bw}$. Furthermore, it is clear that the maximum consistency occurs when $a_{Bj}$ and $a_{jw}$ have the maximum value (equal to $a_{Bw}$), which will conduce to $\xi = 0$. Accordingly, we have:

$$(a_{Bj} - \xi) \times (a_{jw} - \xi) = (a_{Bw} + \xi). \tag{10}$$

As for the maximum inconsistency $a_{Bj} = a_{jw} = a_{Bw}$, we also have:

$$(a_{Bw} - \xi) \times (a_{Bw} - \xi) = (a_{Bw} + \xi). \tag{11}$$

Finally, Equation (11) can be written as:

$$\xi^2 - (1 + 2a_{Bw})\xi + (a_{Bw}^2 - a_{Bw}) = 0. \tag{12}$$

## 3. Numerical Examples

In this section, two numerical examples of SCM in the literature are considered to illustrate the application of our proposed methodology. In SCM, supplier relationship management is divided into three steps: (i) supplier selection, (ii) supplier segmentation, and (iii) supplier development [21]. The first example is a real-world decision-making problem of green supplier selection adopted from Gupta and Barua [49]. The second example is another real-world decision-making problem of supplier segmentation and supplier development adopted from Rezaei [21]. It should be noted that the opinions of 20 experts were taken into account to analyze both numerical examples.

### 3.1. Green Supplier Selection

Green supplier selection is an important issue in producing sustainable products and achieving the goals of green supply chains. Gupta and Barua [49] employed the BWM to prioritize the criteria of green innovation ability for supplier selection problems. To make the appropriate comparisons, we adopt the ranking of the main green innovation criteria mentioned by Gupta and Barua [49] in this research. The seven criteria of green innovation identified by their research are given as follows:

| Criteria | Description |
|---|---|
| C1 | Collaborations |
| C2 | Environmental investments and economic benefits |
| C3 | Resource availability and green competencies |
| C4 | Environmental management initiatives |
| C5 | Research and design initiatives |
| C6 | Green purchasing capabilities |
| C7 | Regulatory obligations, pressures, and market demand |

To discuss the applicability of our proposed methodology, seven decision criteria are identified for the green supplier selection problem (Step 1). Moreover, 20 decision makers are taken into consideration to evaluate these criteria. In Step 2, the decision makers choose the best and worst criteria according to their evaluations. Then, the decision makers perform the pairwise comparisons of best-over-other and other-over-worst using crisp

numbers of 1 to 9. In Step 5, depending on the choices made by decision makers with respect to the best and worst criteria, the grouping of decision makers is conducted. The execution of Steps 1–5 can be seen in Table 2.

**Table 2.** Pairwise comparison vector for the best and worst criteria.

| Panel | DM Group | Best | Worst | P | C1 | C2 | C3 | C4 | C5 | C6 | C7 | DM Num |
|---|---|---|---|---|---|---|---|---|---|---|---|---|
| DM Panel | (1) | C3 | C4 | PB | 2 | 3 | 1 | 9 | 4 | 3 | 2 | (2) |
| | | | | PW | 6 | 4 | 9 | 1 | 3 | 5 | 6 | |
| | | | | PB | 2 | 3 | 1 | 9 | 3 | 2 | 2 | (5) |
| | | | | PW | 7 | 6 | 9 | 1 | 5 | 6 | 7 | |
| | | | | PB | 2 | 2 | 1 | 9 | 5 | 2 | 3 | (6) |
| | | | | PW | 5 | 4 | 9 | 1 | 3 | 6 | 5 | |
| | | | | PB | 2 | 2 | 1 | 9 | 3 | 2 | 3 | (7) |
| | | | | PW | 6 | 4 | 9 | 1 | 3 | 7 | 5 | |
| | | | | PB | 2 | 3 | 1 | 9 | 4 | 2 | 2 | (8) |
| | | | | PW | 6 | 4 | 9 | 1 | 6 | 5 | 8 | |
| | | | | PB | 3 | 2 | 1 | 9 | 4 | 2 | 2 | (10) |
| | | | | PW | 6 | 3 | 9 | 1 | 3 | 6 | 7 | |
| | | | | PB | 2 | 2 | 1 | 8 | 3 | 2 | 2 | (12) |
| | | | | PW | 6 | 4 | 8 | 1 | 3 | 6 | 6 | |
| | | | | PB | 2 | 2 | 1 | 9 | 2 | 2 | 3 | (13) |
| | | | | PW | 5 | 3 | 9 | 1 | 3 | 5 | 4 | |
| | (2) | C3 | C1 | PB | 9 | 3 | 1 | 5 | 3 | 4 | 3 | (3) |
| | | | | PW | 1 | 4 | 9 | 3 | 4 | 3 | 4 | |
| | | | | PB | 8 | 2 | 1 | 5 | 3 | 3 | 2 | (4) |
| | | | | PW | 1 | 5 | 8 | 2 | 3 | 2 | 3 | |
| | | | | PB | 9 | 2 | 1 | 5 | 2 | 5 | 2 | (11) |
| | | | | PW | 1 | 4 | 8 | 2 | 3 | 2 | 5 | |
| | | | | PB | 9 | 3 | 1 | 4 | 4 | 3 | 2 | (17) |
| | | | | PW | 1 | 4 | 9 | 2 | 3 | 4 | 4 | |
| | | | | PB | 9 | 2 | 1 | 5 | 3 | 4 | 2 | (18) |
| | | | | PW | 1 | 4 | 9 | 3 | 5 | 2 | 4 | |
| | | | | PB | 9 | 2 | 1 | 6 | 4 | 4 | 2 | (20) |
| | | | | PW | 1 | 4 | 9 | 2 | 3 | 2 | 4 | |
| | (3) | C7 | C4 | PB | 5 | 2 | 2 | 9 | 3 | 2 | 1 | (14) |
| | | | | PW | 2 | 4 | 4 | 1 | 3 | 5 | 9 | |
| | | | | PB | 5 | 2 | 2 | 9 | 3 | 2 | 1 | (15) |
| | | | | PW | 2 | 4 | 5 | 1 | 3 | 6 | 9 | |
| | | | | PB | 5 | 2 | 2 | 9 | 4 | 2 | 1 | (19) |
| | | | | PW | 3 | 4 | 4 | 1 | 3 | 6 | 9 | |
| | (4) | C7 | C6 | PB | 5 | 3 | 2 | 5 | 3 | 9 | 1 | (1) |
| | | | | PW | 3 | 4 | 5 | 2 | 4 | 1 | 9 | |
| | | | | PB | 5 | 2 | 2 | 5 | 3 | 8 | 1 | (9) |
| | | | | PW | 2 | 3 | 5 | 2 | 3 | 1 | 8 | |
| | | | | PB | 6 | 2 | 2 | 4 | 2 | 9 | 1 | (16) |
| | | | | PW | 2 | 4 | 4 | 3 | 3 | 1 | 9 | |

As can be seen in Table 2, most decision makers chose C3 and C7 as the best criteria. Moreover, C4, C1, and C6 were chosen as the worst criteria. The decision makers are divided into four groups, each of which consists of 8, 5, 4, and 3 decision makers, respectively. The grouping of decision makers into four groups leads to four different models to solve the problem.

After executing pairwise comparisons, the total relative importance of the best criterion over the other criteria (Total $a_{Bj}$) and the total relative importance of all criteria over the worst criterion (Total $a_{jw}$) for each group are calculated using the geometric mean (see Step 6 and Equation (5)). The resulting geometric means for each group can be found in Table 3.

**Table 3.** Total preference of pairwise comparisons for the best and worst criteria.

| Panel | DM Group | Total Best | Total Worst | P | C1 | C2 | C3 | C4 | C5 | C6 | C7 |
|---|---|---|---|---|---|---|---|---|---|---|---|
| DM Panel | (1) | C3 | C4 | PB | 6 | 2 | 1 | 9 | 3 | 6 | 2 |
| | | | | PW | 2 | 4 | 9 | 1 | 3 | 2 | 6 |
| | (2) | C3 | C1 | PB | 9 | 2 | 1 | 5 | 3 | 4 | 2 |
| | | | | PW | 1 | 4 | 9 | 2 | 3 | 2 | 4 |
| | (3) | C7 | C4 | PB | 5 | 2 | 2 | 9 | 3 | 6 | 1 |
| | | | | PW | 2 | 4 | 4 | 1 | 3 | 2 | 9 |
| | (4) | C7 | C6 | PB | 5 | 2 | 2 | 5 | 3 | 9 | 1 |
| | | | | PW | 2 | 4 | 5 | 2 | 3 | 1 | 9 |

It is noteworthy that the calculated values through the geometric mean for all relative importance values are rounded to the nearest number. Now, with regard to the data provided in Table 3 and in order to obtain the optimal value of criteria weights according to Model (7), the following models are derived (Step 7). Models (13)–(16) are utilized to compute the weight of the criteria in Groups (1)–(4), respectively.

DM Group (1):

$$
\text{minimize } \xi \qquad \qquad \text{minimize } k
$$

$$
\text{subject to}
\begin{cases}
|\frac{w_3}{w_1} - 6| \le \xi, \\
|\frac{w_3}{w_2} - 2| \le \xi, \\
|\frac{w_3}{w_4} - 9| \le \xi, \\
|\frac{w_3}{w_5} - 3| \le \xi, \\
|\frac{w_3}{w_6} - 6| \le \xi, \\
|\frac{w_3}{w_7} - 2| \le \xi, \\
|\frac{w_3}{w_4} - 2| \le \xi, \\
|\frac{w_3}{w_4} - 4| \le \xi, \\
|\frac{w_3}{w_4} - 3| \le \xi, \\
|\frac{w_3}{w_4} - 2| \le \xi, \\
|\frac{w_3}{w_4} - 6| \le \xi, \\
\sum_{j=1}^{n} w_j = 1, \\
w_j \ge 0 \text{ for all } j.
\end{cases}
\rightarrow \text{subject to}
\begin{cases}
w_3 - 6w_1 \le kw_1; \quad w_3 - 6w_1 \ge -kw_1; \\
w_3 - 2w_2 \le kw_2; \quad w_3 - 2w_2 \ge -kw_2; \\
w_3 - 9w_4 \le kw_4; \quad w_3 - 9w_4 \ge -kw_4; \\
w_3 - 3w_5 \le kw_5; \quad w_3 - 3w_5 \ge -kw_5; \\
w_3 - 6w_6 \le kw_6; \quad w_3 - 6w_6 \ge -kw_6; \\
w_3 - 2w_7 \le kw_7; \quad w_3 - 2w_7 \ge -kw_7; \\
w_1 - 2w_4 \le kw_4; \quad w_1 - 2w_4 \ge -kw_4; \\
w_2 - 4w_4 \le kw_4; \quad w_2 - 4w_4 \ge -kw_4; \\
w_5 - 3w_4 \le kw_4; \quad w_5 - 3w_4 \ge -kw_4; \\
w_6 - 2w_4 \le kw_4; \quad w_6 - 2w_4 \ge -kw_4; \\
w_7 - 6w_4 \le kw_4; \quad w_7 - 6w_4 \ge -kw_4; \\
w_1 + w_2 + w_3 + w_4 + w_5 + w_6 + w_7 = 1; \\
w_1 \ge 0; w_2 \ge 0; w_3 \ge 0; w_4 \ge 0; \\
w_5 \ge 0; w_6 \ge 0; w_7 \ge 0; \\
k \ge 0
\end{cases}
\tag{13}
$$

DM Group (2):

minimize $\xi$  minimize $k$

$$\text{subject to} \begin{cases} |\frac{w_3}{w_1} - 9| \le \xi, \\ |\frac{w_3}{w_2} - 2| \le \xi, \\ |\frac{w_3}{w_4} - 5| \le \xi, \\ |\frac{w_3}{w_5} - 3| \le \xi, \\ |\frac{w_3}{w_6} - 4| \le \xi, \\ |\frac{w_3}{w_7} - 2| \le \xi, \\ |\frac{w_2}{w_1} - 4| \le \xi, \\ |\frac{w_4}{w_1} - 2| \le \xi, \\ |\frac{w_5}{w_1} - 3| \le \xi, \\ |\frac{w_6}{w_1} - 2| \le \xi, \\ |\frac{w_7}{w_1} - 4| \le \xi, \\ \sum_{j=1}^{n} w_j = 1, \\ w_j \ge 0 \text{ for all } j. \end{cases} \rightarrow \text{subject to} \begin{cases} w_3 - 9w_1 \le kw_1; & w_3 - 9w_1 \ge -kw_1; \\ w_3 - 2w_2 \le kw_2; & w_3 - 2w_2 \ge -kw_2; \\ w_3 - 5w_4 \le kw_4; & w_3 - 5w_4 \ge -kw_4; \\ w_3 - 3w_5 \le kw_5; & w_3 - 3w_5 \ge -kw_5; \\ w_3 - 4w_6 \le kw_6; & w_3 - 4w_6 \ge -kw_6; \\ w_3 - 2w_7 \le kw_7; & w_3 - 2w_7 \ge -kw_7; \\ w_2 - 4w_1 \le kw_1; & w_2 - 4w_1 \ge -kw_1; \\ w_4 - 2w_1 \le kw_1; & w_4 - 2w_1 \ge -kw_1; \\ w_5 - 3w_1 \le kw_1; & w_5 - 3w_1 \ge -kw_1; \\ w_6 - 2w_1 \le kw_1; & w_6 - 2w_1 \ge -kw_1; \\ w_7 - 4w_1 \le kw_1; & w_7 - 4w_1 \ge -kw_1; \\ w_1 + w_2 + w_3 + w_4 + w_5 + w_6 + w_7 = 1; \\ w_1 \ge 0; w_2 \ge 0; w_3 \ge 0; w_4 \ge 0; \\ w_5 \ge 0; w_6 \ge 0; w_7 \ge 0; \\ k \ge 0 \end{cases} \quad (14)$$

DM Group (3):

minimize $\xi$  minimize $k$

$$\text{subject to} \begin{cases} |\frac{w_7}{w_1} - 5| \le \xi, \\ |\frac{w_7}{w_2} - 2| \le \xi, \\ |\frac{w_7}{w_3} - 2| \le \xi, \\ |\frac{w_7}{w_4} - 9| \le \xi, \\ |\frac{w_7}{w_5} - 3| \le \xi, \\ |\frac{w_7}{w_6} - 6| \le \xi, \\ |\frac{w_1}{w_4} - 2| \le \xi, \\ |\frac{w_2}{w_4} - 4| \le \xi, \\ |\frac{w_3}{w_4} - 4| \le \xi, \\ |\frac{w_5}{w_4} - 3| \le \xi, \\ |\frac{w_6}{w_4} - 2| \le \xi, \\ \sum_{j=1}^{n} w_j = 1, \\ w_j \ge 0 \text{ for all } j. \end{cases} \rightarrow \text{subject to} \begin{cases} w_7 - 5w_1 \le kw_1; & w_7 - 5w_1 \ge -kw_1; \\ w_7 - 2w_2 \le kw_2; & w_7 - 2w_2 \ge -kw_2; \\ w_7 - 2w_3 \le kw_3; & w_7 - 2w_3 \ge -kw_3; \\ w_7 - 9w_4 \le kw_4; & w_7 - 9w_4 \ge -kw_4; \\ w_7 - 3w_5 \le kw_5; & w_7 - 3w_5 \ge -kw_5; \\ w_7 - 6w_6 \le kw_6; & w_7 - 6w_6 \ge -kw_6; \\ w_1 - 2w_4 \le kw_4; & w_1 - 2w_4 \ge -kw_4; \\ w_2 - 4w_4 \le kw_4; & w_2 - 4w_4 \ge -kw_4; \\ w_3 - 4w_4 \le kw_4; & w_3 - 4w_4 \ge -kw_4; \\ w_5 - 3w_4 \le kw_4; & w_5 - 3w_4 \ge -kw_4; \\ w_6 - 2w_4 \le kw_4; & w_6 - 2w_4 \ge -kw_4; \\ w_1 + w_2 + w_3 + w_4 + w_5 + w_6 + w_7 = 1; \\ w_1 \ge 0; w_2 \ge 0; w_3 \ge 0; w_4 \ge 0; \\ w_5 \ge 0; w_6 \ge 0; w_7 \ge 0; \\ k \ge 0 \end{cases} \quad (15)$$

DM Group (4):

minimize $\xi$  minimize $k$

$$\text{subject to} \begin{cases} |\frac{w_7}{w_1} - 5| \le \xi, \\ |\frac{w_7}{w_2} - 2| \le \xi, \\ |\frac{w_7}{w_3} - 2| \le \xi, \\ |\frac{w_7}{w_4} - 5| \le \xi, \\ |\frac{w_7}{w_5} - 3| \le \xi, \\ |\frac{w_7}{w_6} - 9| \le \xi, \\ |\frac{w_1}{w_6} - 2| \le \xi, \\ |\frac{w_2}{w_6} - 4| \le \xi, \\ |\frac{w_3}{w_6} - 5| \le \xi, \\ |\frac{w_4}{w_6} - 2| \le \xi, \\ |\frac{w_5}{w_6} - 3| \le \xi, \\ \sum_{j=1}^{n} w_j = 1, \\ w_j \ge 0 \text{ for all } j. \end{cases} \rightarrow \text{subject to} \begin{cases} w_7 - 5w_1 \le kw_1; & w_7 - 5w_1 \ge -kw_1; \\ w_7 - 2w_2 \le kw_2; & w_7 - 2w_2 \ge -kw_2; \\ w_7 - 2w_3 \le kw_3; & w_7 - 2w_3 \ge -kw_3; \\ w_7 - 5w_4 \le kw_4; & w_7 - 5w_4 \ge -kw_4; \\ w_7 - 3w_5 \le kw_5; & w_7 - 3w_5 \ge -kw_5; \\ w_7 - 9w_6 \le kw_6; & w_7 - 9w_6 \ge -kw_6; \\ w_1 - 2w_6 \le kw_6; & w_1 - 2w_6 \ge -kw_6; \\ w_2 - 4w_6 \le kw_6; & w_2 - 4w_6 \ge -kw_6; \\ w_3 - 5w_6 \le kw_6; & w_3 - 5w_6 \ge -kw_6; \\ w_4 - 2w_6 \le kw_6; & w_4 - 2w_6 \ge -kw_6; \\ w_5 - 3w_6 \le kw_6; & w_5 - 3w_3 \ge -kw_6; \\ w_1 + w_2 + w_3 + w_4 + w_5 + w_6 + w_7 = 1; \\ w_1 \ge 0; w_2 \ge 0; w_3 \ge 0; w_4 \ge 0; \\ w_5 \ge 0; w_6 \ge 0; w_7 \ge 0; \\ k \ge 0 \end{cases} \quad (16)$$

The weights obtained for each criterion within each group are shown in Figure 4.

**Figure 4.** Weights of the criteria for each DM group.

The calculated value of $\xi$ for each group is $0.347, 0.154, 0.347$, and $0.154$, respectively (see Table 4). Furthermore, to find the consistency ratio in all groups, we have $a_{BW} = 9$, then, the consistency index for each of the groups is 5.23 (see Table 1). The consistency ratio for the first group is $\frac{0.347}{5.23} = 0.066$. Similarly, the calculated consistency ratios for other groups are $0.029, 0.066, 0.029$, and $0.066$, respectively. It is obvious that the calculated consistency ratio obtained for each group of decision makers is less than 0.1, which indicates that the output results are highly consistent.

**Table 4.** Weight of the criteria.

| Panel | DM Group | $W_1$ | $W_2$ | $W_3$ | $W_4$ | $W_5$ | $W_6$ | $W_7$ | $\xi$ | N |
|-------|----------|-------|-------|-------|-------|-------|-------|-------|-------|---|
| DM Panel | (1) | 0.063 | 0.153 | 0.358 | 0.038 | 0.107 | 0.063 | 0.216 | 0.347 | 8 |
| | (2) | 0.039 | 0.164 | 0.352 | 0.073 | 0.124 | 0.085 | 0.163 | 0.154 | 6 |
| | (3) | 0.071 | 0.163 | 0.163 | 0.041 | 0.114 | 0.067 | 0.381 | 0.347 | 3 |
| | (4) | 0.072 | 0.163 | 0.19 | 0.072 | 0.112 | 0.039 | 0.351 | 0.154 | 3 |
| W | Sum Weight | 1.167 | 3.186 | 6.035 | 1.081 | 2.278 | 1.449 | 4.916 | - | 20 |
| | Final Weight | 0.058 | 0.159 | 0.301 | 0.054 | 0.113 | 0.072 | 0.245 | - | 1 |

As can be seen in Table 4, due to the unequal number of decision makers in each group, the impact of the number of decision makers in each group should be taken into consideration to calculate the final optimal weight of each criterion. Therefore, we can find the final criteria weights considering the weights obtained and the number of decision makers in each group (Step 8).

Figure 5 shows the priorities of the green innovation criteria for supplier selection considering the decision makers' opinions.

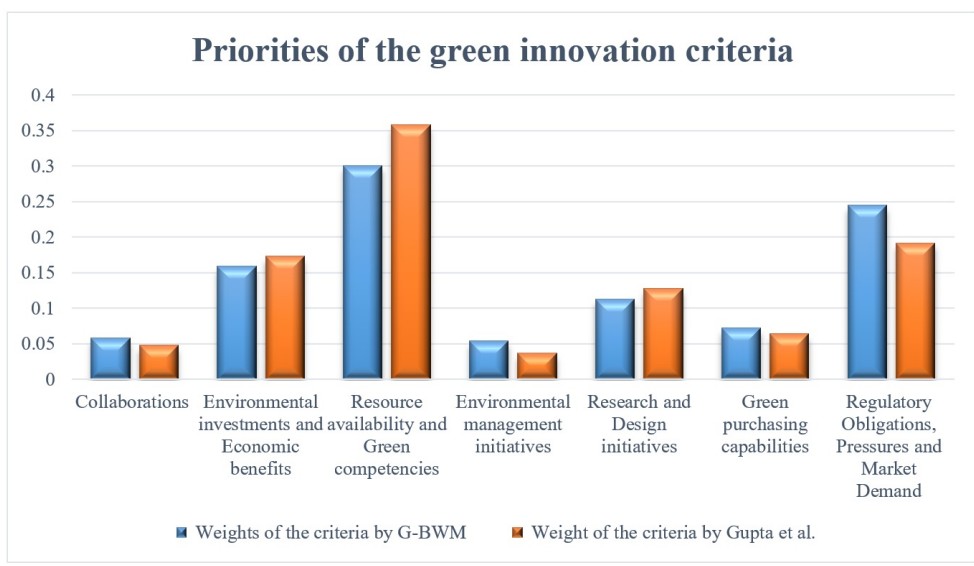

**Figure 5.** Priorities of the green innovation criteria.

The prioritization result of the green innovation criteria for supplier selection by the G-BWM is exactly the same as the result obtained by Gupta and Barua [49].

$$G\text{-}BWM: \quad C3 > C7 > C2 > C5 > C6 > C1 > C4,$$
$$Gupta\ and\ Barua: \quad C3 > C7 > C2 > C5 > C6 > C1 > C4.$$

### 3.2. Linking Supplier Development to Supplier Segmentation

In Rezaei et al. [23], a high-tech Chinese company specializing in the field of testing instruments is used. There are two sets of criteria according to capability and willingness. These selected criteria were investigated as follows:

**Capability**
$C_1^c$: Technical capability
$C_2^c$: Product quality capability
$C_3^c$: Delivery capability
$C_4^c$: Intangible capability
$C_5^c$: Service capability
$C_6^c$: Financial/cost capability
$C_7^c$: Sustainable capability
$C_8^c$: Organizational capability

**Willingness**
$C_1^w$: Willingness to improve performance
$C_2^w$: Willingness to share information
$C_3^w$: Willingness to rely on each other
$C_4^w$: Willingness to get involved in long-term relationship

Rezaei et al. [23] evaluated these criteria using BWM and obtained their weights. Here, we employ our proposed G-BWM with the help of 20 decision makers. After identifying the mentioned criteria (Step 1), the 20 decision makers select the best and worst criteria for each criterion related to capability and willingness (Step 2). Then, each decision maker performs the pairwise comparisons between best-over-other criteria and other-over-worst criterion using the crisp numbers of 1 to 9 (Steps 3 and 4). In Step 5, the decision makers are divided into different groups based on the choices of the best and worst criteria. The results of the execution of Steps 1–5 for willingness and capability criteria are given in Tables 5 and 6. As can be seen in Table 5, the decision makers are divided into two groups in order to evaluate the willingness criteria. Thirteen decision-makers selected C1 as the best criterion, and seven decision-makers selected C4 as the best one. Moreover, all decision makers in both groups selected C2 as the worst criterion.

**Table 5.** Pairwise comparison vector for the best and worst willingness criteria.

| Panel | DM Group | Best | Worst | P | C1 | C2 | C3 | C4 | DM Num |
|-------|----------|------|-------|-----|----|----|----|----|--------|
| DM Panel | (1) | C1 | C2 | PB | 1 | 9 | 3 | 2 | (1) |
| | | | | PW | 9 | 1 | 5 | 5 | |
| | | | | PB | 1 | 9 | 3 | 2 | (2) |
| | | | | PW | 9 | 1 | 4 | 6 | |
| | | | | PB | 1 | 9 | 4 | 2 | (3) |
| | | | | PW | 9 | 1 | 4 | 4 | |
| | | | | PB | 1 | 9 | 3 | 3 | (5) |
| | | | | PW | 9 | 1 | 4 | 5 | |
| | | | | PB | 1 | 9 | 3 | 2 | (6) |
| | | | | PW | 9 | 1 | 4 | 5 | |
| | | | | PB | 1 | 8 | 4 | 3 | (9) |
| | | | | PW | 8 | 1 | 5 | 6 | |
| | | | | PB | 1 | 8 | 3 | 2 | (11) |
| | | | | PW | 8 | 1 | 5 | 5 | |
| | | | | PB | 1 | 8 | 3 | 2 | (12) |
| | | | | PW | 8 | 1 | 4 | 5 | |
| | | | | PB | 1 | 9 | 4 | 2 | (13) |
| | | | | PW | 9 | 1 | 3 | 4 | |
| | | | | PB | 1 | 9 | 3 | 2 | (14) |
| | | | | PW | 9 | 1 | 4 | 5 | |
| | | | | PB | 1 | 9 | 3 | 2 | (17) |
| | | | | PW | 9 | 1 | 3 | 4 | |
| | | | | PB | 1 | 9 | 4 | 3 | (18) |
| | | | | PW | 9 | 1 | 3 | 5 | |
| | | | | PB | 1 | 9 | 3 | 2 | (20) |
| | | | | PW | 9 | 1 | 5 | 4 | |
| | (2) | C4 | C2 | PB | 2 | 1 | 3 | 8 | (4) |
| | | | | PW | 4 | 8 | 4 | 1 | |
| | | | | PB | 2 | 1 | 3 | 9 | (7) |
| | | | | PW | 5 | 9 | 5 | 1 | |
| | | | | PB | 3 | 1 | 4 | 9 | (8) |
| | | | | PW | 3 | 9 | 4 | 1 | |
| | | | | PB | 2 | 1 | 3 | 9 | (10) |
| | | | | PW | 5 | 9 | 5 | 1 | |
| | | | | PB | 2 | 1 | 3 | 9 | (15) |
| | | | | PW | 4 | 9 | 4 | 1 | |
| | | | | PB | 3 | 1 | 4 | 9 | (16) |
| | | | | PW | 4 | 9 | 5 | 1 | |
| | | | | PB | 2 | 1 | 3 | 9 | (19) |
| | | | | PW | 4 | 9 | 4 | 1 | |

**Table 6.** Pairwise comparison vector for the best and worst capability criteria.

| Panel | DM Group | Best | Worst | P | C1 | C2 | C3 | C4 | C5 | C6 | C7 | C8 | DM Num |
|---|---|---|---|---|---|---|---|---|---|---|---|---|---|
| DM Panel | (1) | C2 | C8 | PB | 7 | 1 | 2 | 8 | 4 | 3 | 4 | 9 | (1) |
| | | | | PW | 2 | 9 | 8 | 2 | 5 | 6 | 5 | 1 | |
| | | | | PB | 6 | 1 | 3 | 7 | 5 | 4 | 4 | 9 | (6) |
| | | | | PW | 3 | 9 | 6 | 2 | 3 | 5 | 4 | 1 | |
| | | | | PB | 7 | 1 | 2 | 7 | 6 | 3 | 4 | 9 | (7) |
| | | | | PW | 2 | 9 | 7 | 2 | 3 | 7 | 6 | 1 | |
| | | | | PB | 7 | 1 | 2 | 7 | 5 | 4 | 4 | 8 | (12) |
| | | | | PW | 3 | 8 | 8 | 3 | 3 | 4 | 5 | 1 | |
| | | | | PB | 6 | 1 | 3 | 8 | 4 | 3 | 4 | 9 | (17) |
| | | | | PW | 4 | 9 | 6 | 2 | 4 | 7 | 6 | 1 | |
| | | | | PB | 5 | 1 | 2 | 7 | 5 | 3 | 4 | 9 | (18) |
| | | | | PW | 3 | 9 | 7 | 2 | 3 | 6 | 5 | 1 | |
| | (2) | C2 | C4 | PB | 7 | 1 | 2 | 9 | 6 | 4 | 5 | 8 | (2) |
| | | | | PW | 3 | 9 | 8 | 1 | 4 | 4 | 4 | 2 | |
| | | | | PB | 7 | 1 | 2 | 8 | 5 | 3 | 6 | 7 | (4) |
| | | | | PW | 3 | 8 | 7 | 1 | 3 | 7 | 4 | 2 | |
| | | | | PB | 6 | 1 | 2 | 9 | 4 | 3 | 5 | 8 | (10) |
| | | | | PW | 4 | 9 | 8 | 1 | 5 | 5 | 3 | 2 | |
| | | | | PB | 6 | 1 | 2 | 8 | 5 | 2 | 5 | 7 | (11) |
| | | | | PW | 3 | 8 | 7 | 1 | 4 | 7 | 4 | 2 | |
| | | | | PB | 7 | 1 | 2 | 9 | 5 | 3 | 4 | 7 | (13) |
| | | | | PW | 2 | 9 | 9 | 1 | 3 | 4 | 4 | 3 | |
| | | | | PB | 8 | 1 | 3 | 9 | 5 | 3 | 5 | 7 | (20) |
| | | | | PW | 2 | 9 | 8 | 1 | 4 | 5 | 4 | 3 | |
| | (3) | C3 | C8 | PB | 7 | 2 | 1 | 8 | 4 | 3 | 4 | 9 | (14) |
| | | | | PW | 3 | 8 | 9 | 2 | 4 | 6 | 5 | 1 | |
| | | | | PB | 6 | 3 | 1 | 7 | 4 | 4 | 3 | 9 | (15) |
| | | | | PW | 3 | 6 | 9 | 3 | 5 | 5 | 4 | 1 | |
| | | | | PB | 7 | 2 | 1 | 8 | 5 | 3 | 4 | 9 | (19) |
| | | | | PW | 3 | 7 | 9 | 2 | 4 | 6 | 5 | 1 | |
| | | | | PB | 7 | 2 | 1 | 8 | 4 | 3 | 4 | 2 | (12) |
| | | | | PW | 3 | 8 | 2 | 2 | 3 | 7 | 5 | 1 | |
| | | | | PB | 8 | 2 | 1 | 9 | 4 | 3 | 4 | 9 | (13) |
| | | | | PW | 2 | 7 | 9 | 1 | 4 | 6 | 6 | 1 | |
| | (4) | C3 | C4 | PB | 6 | 2 | 1 | 9 | 5 | 4 | 4 | 8 | (3) |
| | | | | PW | 4 | 8 | 9 | 1 | 4 | 6 | 5 | 2 | |
| | | | | PB | 5 | 3 | 1 | 8 | 6 | 3 | 4 | 7 | (5) |
| | | | | PW | 4 | 7 | 8 | 1 | 4 | 7 | 5 | 3 | |
| | | | | PB | 6 | 2 | 1 | 9 | 5 | 3 | 5 | 8 | (16) |
| | | | | PW | 3 | 8 | 9 | 1 | 4 | 6 | 5 | 2 | |

Unlike the willingness criteria, the decision makers are divided into four groups in order to evaluate the capability criteria. The main reason is that there are more criteria in this set. By increasing the number of criteria, the contradictions also increase in selecting the best and worst criteria. As can be seen in Table 6, the two criteria of C2 and C3 were selected as the best criteria, and the two criteria of C4 and C8 were selected as the worst ones.

By executing Steps 6–8, the final weights for each of the capability and willingness criteria are calculated. Figure 6 represents the priorities of the willingness and capability criteria for the supplier development/segmentation problem considering the decision makers' opinions based on our proposed G-BWM and BWM suggested by Rezaei et al. [23].

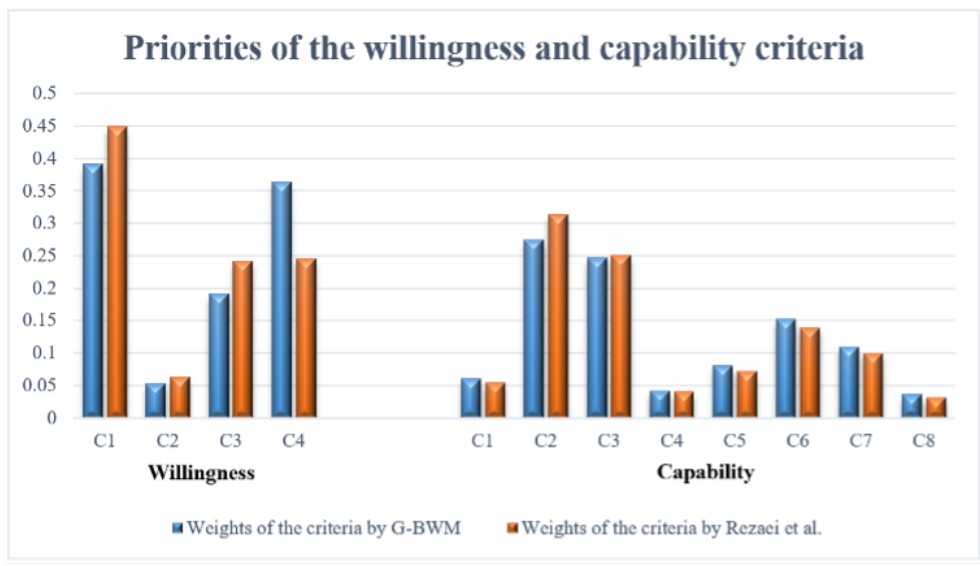

**Figure 6.** Priorities of the willingness and capability criteria.

As can be seen in Figure 6, all priorities of willingness and capability criteria obtained using G-BWM and BWM are the same.

## 4. Comparative Analysis and Discussion

In AHP, each criterion must be compared with all the other criteria in order to determine the criteria weights. So, for $n$ criteria, we need to execute $n^2$ pairwise comparisons. Due to the equality preference of each criterion to itself, $n$ comparisons are reduced accordingly. Moreover, half of the values in the pairwise comparison matrix are written in reverse, and at least $n(n-1)/2$ pairwise comparisons need to be executed by decision makers.

BWM was suggested to deal with the challenges of AHP in pairwise comparisons and inconsistency issues. Rezaei [22] stated that the main cause of inconsistency is an unreasonable method in executing pairwise comparisons. Accordingly, he could reduce the number of pairwise comparisons to $2n - 3$ in order to identify the weight of $n$ criteria by dividing the steps of pairwise comparisons into two parts: reference comparison and secondary comparisons [22].

Mi et al. [50] compared BWM and AHP to show the difference in the number of pairwise comparisons. Figure 7 represents the difference in the required number of pairwise comparisons between the BWM and AHP method. The $x$-axis shows the number of objects to be compared in the decision-making process and the $y$-axis displays the least number of pairwise comparisons executed in each method to find the weights of compared objects. When increasing the number of objects, the number of pairwise comparisons needed by BWM grows linearly (blue points) while the number of pairwise comparisons required by AHP increases exponentially (orange points) [50].

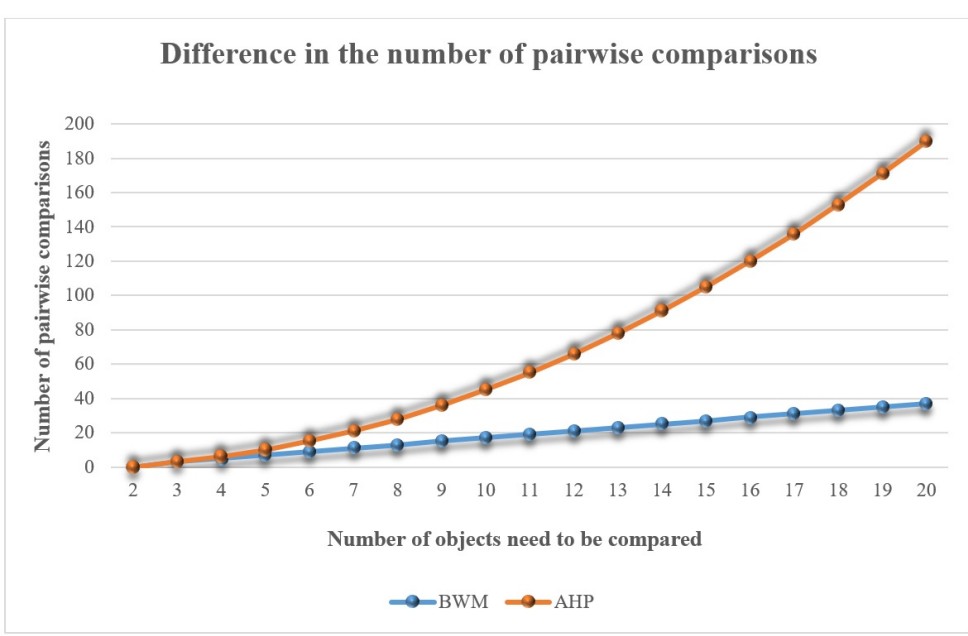

**Figure 7.** Differences in the number of pairwise comparisons needed by BWM and AHP.

In BWM, a mathematical model is needed to obtain the criteria weights according to decision maker needs. For *n* decision makers, it is also required to derive *n* mathematical models. The previously developed models for group decision making based on BWM [44,45] have some difficulties that can restrict their applications. By growing the size of the decision maker panel, the scale of the mathematical models increases linearly. Due to this, many examples given by these previous models only used three or four decision makers. In other words, increasing the number of decision makers limits the applications of the models.

Our proposed G-BWM tries to group decision makers based on their opinions and resolve this drawback. Decision makers are divided into different groups based on the selection of similar best and worst criteria. This kind of grouping of decision makers reduces the scale of the mathematical model. To demonstrate the different mathematical models needed by BWM and our proposed G-BWM, a group decision-making process with seven criteria is compared. According to these seven criteria, we asked 30 experts to fill in pairwise comparison questionnaires based on the BWM framework. The obtained results for comparing the required number of mathematical models in BWM and G-BWM are interesting.

Figure 8 shows the difference between the required number of mathematical models to determine the criteria weights in BWM and our proposed G-BWM. The x-axis represents the number of decision makers to be compared in the group decision-making process. The y-axis indicates the least number of mathematical models required to obtain the criteria weights in each method. When increasing the number of decision makers, the number of mathematical models needed by BWM (and other developed models for group decision making based on BWM) to obtain the criteria weights grows linearly (orange points) while the number of mathematical models required by our proposed G-BWM to obtain the criteria weights follows a step-by-step upward trend (blue points). This difference is much greater for a large number of decision makers. Figure 8 depicts the superior performance of the suggested G-BWM approach against the BWM and other BWM-based group decision-making models.

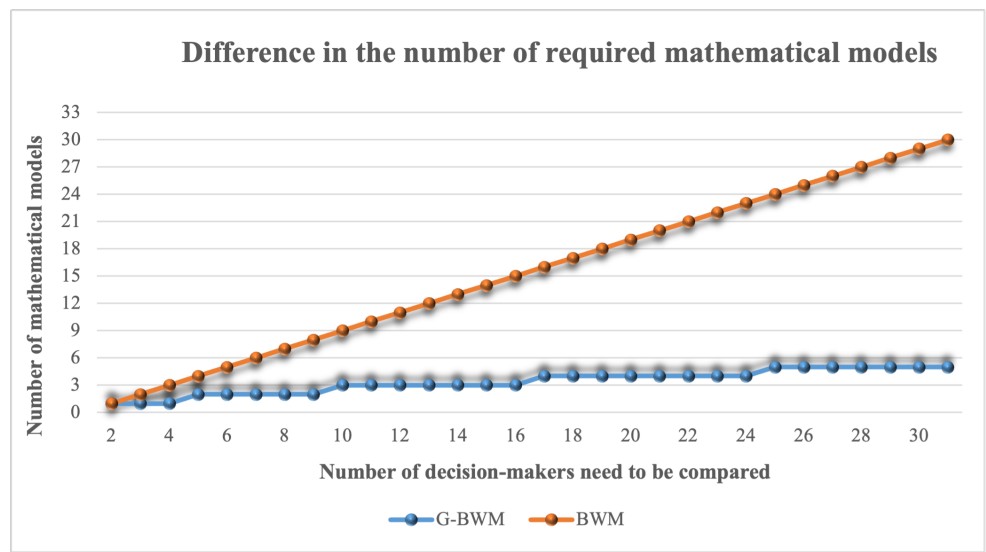

**Figure 8.** Differences in the number of required mathematical models.

In the group decision-making process, conflicts may occur among the opinions of decision makers. In the previously developed models for group decision making based on BWM, the senior decision maker eliminated the inconsistent opinions of some decision makers. Safarzadeh et al. [45] asserted that senior decision makers can determine the best and worst criteria at the first step to be regarded for the final decision. Therefore, assume that there is a group of decision makers to obtain the best and worst criteria. According to the model offered by Safarzadeh et al. [45], if 49% of decision makers select criterion A as the best and 51% of decision makers select criterion B as the best, the senior decision maker prefers to eliminate the opinions of the decision makers who selected criterion A and considers criterion B as the best criterion for the final decision. In fact, autocratic decision making rarely incorporates the opinions of all decision makers. In our proposed G-BWM, the final decision is based on the collective opinions of the decision makers. The decision makers are divided into different groups based on their opinions, and without any elimination, the opinions of each group are assessed.

## 5. Conclusions and Outlook

This study developed a novel approach for group decision making based on BWM, called G-BWM. The proposed approach has a hierarchical framework in which decision makers are grouped according to their opinions after providing the required evaluations for the relative importance of criteria. Moreover, using the BWM structure, the weights obtained for each group are computed and merged to obtain the final criteria weights as the optimal weights. The proposed G-BWM is vector based and is much easier to use and more efficient than matrix-based MCDM techniques such as AHP. To validate the applicability of the G-BWM, two numerical examples were adopted from the literature for the group decision making in SCM. The aim was to demonstrate how the analyzer can use the G-BWM and check the performance and compliance. The results revealed that the proposed G-BWM has a high consistency ratio and reliability. In other words, our suggested G-BWM has several distinctive features that make it an interesting and robust method, which can be described as follows:

i.   The G-BWM can be utilized individually to obtain the criteria weights and it can be also be hybridized with other MCDM methods to do so,

ii.  In the previous approaches offered for group decision making based on BWM, the scale of mathematical models increases with the size of the decision maker panel. Our proposed G-BWM is based on democratic decision making and reduces the scale of the mathematical model by grouping the decision makers based on their opinions on choosing the best and worst criteria,

iii.  In case of conflicts among decision makers, the proposed G-BWM keeps them in different groups based on the similarity of their opinions instead of eliminating the opinions of decision makers who have a low level of expertise.

As the main weakness or limitation of the G-BWM, it should be noted that it cannot provide a self-acting tuning process when the consistency ratio is undesirable. To resolve this issue, a special framework can be employed to control the relative importance values assigned by decision makers for pairwise comparisons. Moreover, different weighting methods such as equivalent and priority criteria [51] can be considered compared to the proposed one. Finally, to make the proposed approach closer to real-world scenarios, the best–worst scaling (BWS) technique [52] can be employed to deal with the estimation of choice probability.

**Author Contributions:** G.H.: conceptualization, methodology, project administration, writing—original draft, investigation; R.S.: formal analysis, software; J.W.: writing—review and editing, validation; H.T.: investigation, visualization, validation; E.B.T.: data curation, supervision, writing—review and editing. All authors have read and agreed to the published version of the manuscript.

**Funding:** The APC was funded by FIM UHK Excellence Project 2021: Decision Support Systems: Principles and Applications 3.

**Institutional Review Board Statement:** Not applicable.

**Informed Consent Statement:** Not applicable.

**Data Availability Statement:** All the required data are included in the manuscript.

**Acknowledgments:** The project was supported by FIM UHK Excellence Project 2021: Decision Support Systems: Principles and Applications 3.

**Conflicts of Interest:** The authors declare no conflict of interest.

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
