# Peer review of "A Novel Approach for Group Decision Making Based on the Best–Worst Method (G-BWM): Application to Supply Chain Management"

_mathematics, doi:10.3390/math9161881_

Round 1

Reviewer 1 Report

Dear Authors,

Thank you for producing this manuscript which is clearly conceptualised and has a simple yet appropriate methodology whereby a comparison between two methods of decision making is illustrated and analysed.

The problem is well chosen as it is commonly found in applied decision making. The novel G-BWM process is succintly represented with precise mathematical epressions and anappropriate process flowchart. The visual methods used for the comparative analysis are well chosen. 

The analysis and discussion highlights appropriate differences bewteen existing processes for group decsion making and the proposed G-BWM process.

The conclusions and opportunities for further research are justified and appropriate.

The most sigificant and recurring problem in this manuscript is the cumbersome phraseology used throughout. Errors are are so prevalent as to be present in most sentences. Whilst it is easily possible to decypher the meaning such that a logical progression of ideas is revealed, the language is not in keeping with a peer reviewed journal of this stature. 

Specific examples of errors in language in the abstract alone are:

  1. The first sentence of the abstract.
  2. Irrelevant fact in sentence 3 of the abstract.
  3. Use of the phrase "simple usage" in line 12 where "ease of use" more accurately conveys the intended meaning.

To indicate how prevalent the errors are, I have also reviewed language errors from line 100 through to 110:

  1. Gramatical error in line 100.
  2. Tautology in line 103.
  3. Gramatical error in the sentence begining in line 107.
  4. Tautology in the sentence beginning in line 109.

Whilst the errors are prevalent, they are not difficult to correct, hence I have suggested this is a minor revision. I encourage you to seek expert assistance with the language used in this manuscript.

Kind regards.

Reviewer 2 Report

The paper presents a new group decision making method. The main characteristic of the proposal is that it groups several decision makers beforehand based on their most and least preferred criteria. Being sincere, I feel that the added value to the field of MCDM is little, especially bearing in mind that nowadays most of the complexity problems for building MCDM models come from the cost of collecting the information rather than from the cost of computing the model itself. In the following I provide some additional comments:

A scale from 1 to 9 is used for pairwise comparisons, but this scale is typically used for absolute evaluations. Although this is clearly not a big problem, it is definitely not standard.

A phrase states that “The geometric mean is the same as the arithmetic mean”. This is obviously not correct.

Other options than the geometric mean could be considered (see, e.g., M. Grabisch, J.L. Marichal, R. Mesiar, E. Pap, Aggregation functions: Means, Information Sciences 181 (2011) 1—22).

Closely related, but in the context of rankings, the following article might be of interest: J.L. García-Lapresta, A.A.J., Marley, M. Martínez-Panero, Characterizing best-worst voting systems in the scoring context 34 (2010) 487—496.

Perhaps one may find more elaborate ways of grouping the decision makers. For instance, I am think of some clustering techniques.

The interpretation of the weights (appearing, e.g. in Step 7 of the algorithm in page 5) might be provided.

Subsection 2.2 discusses the notion of “consistency”. However, there are multiple different definitions of this concept in the literature. To which one do you refer?

The English writing should be further improved.

Minor comments. Table 3, a parenthesis should be removed at the fourth row and third column. Line 194 n2 should be n^2.

Reviewer 3 Report

Authors investigate a problem of "Title:   A novel approach for the group decision-making based on the best-worst method (G-BWM): Application to green supplier selection".

The work is aimed at illustrating group decision-making, when there is a large number of decision-makers.  The authors wanted to show how different decision-makers with varying degrees of importance can be classified for optimal analysis and how to obtain optimal weight criteria without eliminating the opinions of decision makers who have a minority.

As a mathematical support, the method of multi-criteria decision-making (MCDM) and the introduction of the best-worst method (BWM) is used, which improves the consistency coefficient by performing fewer paired comparisons.

The authors reviewed the literature during the study.

The best - worst (BWM) method is presented in greater detail, the boiler is finalized by the authors and is called G-BWM.

Based on the review, an approach based on the opinion of a group of qualified experts was chosen, i.e. decision-making in conditions of uncertainty.

But first, we will give a description of decision-making under conditions of certainty.

When organizing and developing of the agricultural production, strategic goals are developed, on the basis of which business goals and plans of the agricultural production.

Financial statements are formed, as well as the quality (return on assets) and efficiency (period of return of capital) of investments, which directly influence the financial performance of corporations, are assessed.

The movement of organizational, technological, accounting, and financial information from suppliers to consumers and vice versa represents a fragment of the "Digital Economy", which is the basis for making managerial decisions.

But the authors do not perceive (or do not know) the Digital Economy for management decisions, including audit.

Therefore, the transition to fuzzy information and its processing seems highly questionable.

Fuzzy information can be used successfully, for example, in the analysis of competitive firms.

Let's pass to mathematics.

  1. Section 3 states: "Of all the criteria, the authors chose seven main criteria:

C1 Collaborations;  C2 Environmental investments and economic benefits; ...  

C7 Regulatory obligations, pressures and market demand»

Question: "Why seven criteria and not five or 10 elements?" There is no explanation.

  1. Section 3 states:«In Step 2, the decision-makers choose the best and worst criteria according to their evaluations. Then, the decision-makers perform the pairwise comparisons of best-over-other and other-over-worst using crisp numbers of 1 to 9».

Question: "What do the indicators mean: [1, 2, … , 9] in physical units: Sales volume; environmental costs; profits ..., on the basis of which decisions are made, and not by weight factors." There is no explanation.

  1. Using the previously presented methods (MCDM, BWM, G-BWM), page 10 states: "The weights obtained for each criterion in each group are shown in Figure 4", which are presented in Table 4.  Weight of the criteria

But the authors do not understand the influence of "weighting factors" on the result of the decision (as well as the authors of literary references).

The authors suggest that the weighting factors equalize the criteria for making decisions. The more weight, the more important (priority) the corresponding criterion.

 For example, from table 4

  1. W1 weight=0,063=0,06   (rounded)     

W2 weight=0,153=0,15   (rounded)     

it follows that   W2=0,15 is more important (priority)  than   W1=0,06.

Let's analyse the result mathematically.

The value of the first indicator  W1 indicates "a" and the value of the second indicator  W2 indicates "b." The work assumes that the weights equalize their effect on the task. Then the relation is carried out

W1*a =  W2*b  or

0.06*a=0.15*b

From here

a=0.15*b/0.06 or

a=2.5*b.

As a result, we learn that the  W1 criterion is 2.5 times more effective than the  W2 criterion, i.e. the result with accuracy vice versa.

In work https://rdcu.be/bhZ8i in the application

I conducted a theoretical analysis of the use of weights and showed that they give quite the wrong result that the author puts into them.

Therefore, all further arithmetic actions will lead to the answer that the authors need.

Таким образом, в целом рецензия на статью носит негативный характер.

Рекомендации.

В то же время я считаю, что статья должна быть напечатана в вашем журнале (вместе с рецензией).

Объем работы показывает, что авторы работали над статьей. В науке (и особенно в математике) отрицательный результат также является результатом. Вполне возможно, что я не увидел чего-то, что заинтересует читателей.

Поэтому я рекомендую эту работу опубликовать вместе с рецензией.

Reviewer 4 Report

The problem considered in the article (multidimensional decision making) is well known and was discussed in literature for decades. It is logical to expect something really new there, but it is not what presented. The only addition to existing procedures (which are themselves very questionable for many reasons) is that authors propose to break experts in groups instead of judging them all together. But this is a very general statistical recommendation, to be applied to any data, when it's possible to make such a classification. The next step is, however, what to do with those groups, what is this grouping for? If, for example, grouping the people to males and females allows to make better medicine for each gender - it is very useful and meaningful. But if the group of 20 experts is broken into 4 classes and each has its own preferences - what to do with those preferences, how to apply them to the problem of question? Authors in fact do not address this question.

Look at the Table 4. Weight for group 1 for W4 is 0.038, while in a group 2 - 0.073, two times higher. And what? The final weight, 0.054, is just a function of sum of all weights, irrespective of the groups difference.  What those groups then are needed for? 

Other problems are the following.

  1. What is the logic of the grouping? In fact, it is a clustering problem, but no algorithm of division was discussed or I missed it. No metric was proposed. What if two experts are close to each other by W, but far away for B, etc.? 
  2. Authors do not touch the most controversial aspect of MC estimation - the presence of non-transitional relationships between the scores of one expert. Table 2 provides strange impression on that and other matters. Why it is caller "pairwise", while it doesn't have pairwise comparisons (each criteria with all others or each expert with all others)? It seems that each expert gave in fact just ranking off all C, instead of just saying - this is the best of 7, and this is the worst. If that, why, say, expert 2 gives TWO rankings - for W and for B, two lines in a table? It is unclear.  
  3.  There is a big theory of WB estimations, where some underlying models, not just estimations per se are considered (see references here, for example: Stan Lipovetsky (2020) Express analysis for prioritization: Best–Worst
    Scaling alteration to System 1, Journal of Management Analytics, 7:1, 12-27, DOI:
    10.1080/23270012.2019.1702112) Why authors didn't even refer to that stuff, which looks very relevant to their problem?
  4. On the other hand, authors stress out elementary and obvious things  like those on figures 6 and 7, which do not need such elaboration. 

Considering all that, I would not recommend it for publication, unless it will be seriously redone with all explanations needed.

Round 2

Reviewer 2 Report

The main characteristic of the proposed group decision making method is that it groups several decision makers beforehand based on their most and least preferred criteria. As mentioned in my original review, I feel that the added value to the field of MCDM is little, especially bearing in mind that nowadays most of the complexity problems for building MCDM models come from the cost of collecting the information rather than from the cost of computing the model itself. The authors do not even comment on this on their response to reviewers.

I am aware that some existing works (e.g., Linking supplier development to supplier segmentation using Best Worst Method by J. Rezaei, J. Wang and L. Tavasszy) also deal with this problem. It could be interesting to explain the main differences with said paper.

Also, I do not quite understand the problem with clustering techniques that is mentioned in the response to reviewers.

Overall, I think a smaller and more-specific venue would be a better fit for this article.

Reviewer 4 Report

It's corrected

Author Response

We are very thankful to the respected reviewer for his/her careful review and comments.